# Selenium Nanoparticles (Se-NPs) Alleviates Salinity Damages and Improves Phytochemical Characteristics of Pineapple Mint (*Mentha suaveolens* Ehrh.)

**DOI:** 10.3390/plants11101384

**Published:** 2022-05-23

**Authors:** Fatemeh Kiumarzi, Mohammad Reza Morshedloo, Seyed Morteza Zahedi, Hasan Mumivand, Farhad Behtash, Christophe Hano, Jen-Tsung Chen, Jose M. Lorenzo

**Affiliations:** 1Department of Horticultural Science, Faculty of Agriculture, University of Maragheh, Maragheh 83111-55181, Iran; kiumarzi.faty@gmail.com (F.K.); s.m.zahedi@maragheh.ac.ir (S.M.Z.); behtash@maragheh.ac.ir (F.B.); 2Department of Horticultural Science, Faculty of Agriculture, Lorestan University, Khorramabad 68151-44316, Iran; hmumivand@ut.ac.ir; 3Laboratoire de Biologie des Ligneux et des Grandes Cultures, INRA USC1328, Orleans University, CEDEX 2, 45067 Orléans, France; hano@univ-orleans.fr; 4Department of Life Sciences, National University of Kaohsiung, Nanzih, Kaohsiung 811, Taiwan; 5Centro Tecnológico de la Carne de Galicia, Rúa Galicia N° 4, Parque Tecnológico de Galicia, San Cibraodas Viñas, 32900 Ourense, Spain; 6Área de Tecnología de los Alimentos, Facultad de Ciencias de Ourense, Universidad de Vigo, 32004 Ourense, Spain

**Keywords:** piperitenone oxide, abiotic stress, medicinal plants, active ingredient, nanoparticles

## Abstract

The present study examined the effects of foliar spray of selenium nanoparticles (0, 10 and 20 mg/L) on the yield, phytochemicals and essential oil content and composition of pineapple mint (*Mentha suaveolens* Ehrh.) under salinity stress (0, 30, 60 and 90 mM NaCl). Obtained results demonstrated that severe salinity stress reduced the fresh weight (FW) and plant height (PH) by 16.40% and 19.10%, respectively compared with normal growth condition. On the other hands, under sever salinity stress relative water content (RWC) and chlorophyll index were reduced by 18.05% and 3.50%, respectively. Interestingly, selenium nanoparticles (Se-NPs; 10 mg/L) application improved the pineapple mint growth. Based on GC-FID and GC-MS analysis, 19 compounds were identified in pineapple mint essential oil. Foliar application of Se-NPs and salinity did not change the essential oil content of pineapple mint, however, the essential oil compounds were significantly affected by salinity and Se-NPs- applications. Foliar application of Se-NPs- had a significant effect on piperitenone oxide, limonene, jasmone, viridiflorol and *β*-myrsene under different salinity levels. The highest percentage of piperitenone oxide (79.4%) as the major essential oil component was recorded in the no salinity treatment by applying 10 mg/L of nanoparticle. Interestingly, application of 10 mg L^−1^ Se-NPs- under 60 mM NaCl increased the piperitenone oxide content by 9.1% compared with non-sprayed plants. Finally, the obtained results demonstrated that foliar application of Se-NPs (10 mg L^−1^) can improve the pineapple mint growth and secondary metabolites profile under saline conditions.

## 1. Introduction

*M. suaveolens*, Ehrh., also known as pineapple mint is a perennial herbaceous medicinal herb that belongs to the Lamiaceae family [1]. This species is native to southern and western Europe and is cultivated in some parts of Europe [2]. Pineapple mint is one of the variegated cultivars of *M. suaveolens* possessing bumpy and hairy leaves usually surrounded with white margins [3]. The plant has many therapeutic effects, including analgesic, antispasmodic, sedative, appetizing, stimulating, tonic, anti-viral and anti-convulsive [2,4]. Also, it is useful in treating digestive problems and respiratory ailments [5]. Pineapple mint essential oil is a rich source of piperitenone oxide with insecticidal, antibacterial and antifungal properties [2,6]. The amount and type of compounds in pineapple mint essential oil is different, however, research on the essential oils collected from its different ecotypes shows a high percentage of oxygenate monoterpenes such as piperitone oxide and piperitenone oxide [7,8]. Other groups of monoterpenes such as piperitenone, pulegone, piperitone dihydrocarvone, and menthol have been reported from pineapple mint essential oil [5,6,9]. Plants synthesize several different kinds of secondary metabolites during their evolutionary period to cope with environmental conditions [10,11,12].

Salinity by affecting the physiological and biochemical parameters limit the plant growth and yield [13,14,15]. Osmotic stress, Ion toxicity and reactive oxygen species (ROS) production are the main effects of salinity, decreasing the plant growth and photosynthesis [16]. Interestingly, plants developed an antioxidant system (enzymatic and non-enzymatic antioxidants) to prevent oxidative damages [17,18,19]. Therefore, unlike all agricultural products that are damaged in terms of production in stressful conditions, the amounts of secondary metabolites such as phenols, terpenes, and nitrogenous compounds may increase in slight stress conditions and as a result have higher economic returns [20,21]. However, the production of secondary metabolites in response to salinity is depends on plants species, class of metabolites and other factors, for instance, the production of tropane alkaloids in pricklyburr (*Datura inoxia*) increased under salinity stress [20]. On the other hands, in some medicinal plants basil and fennel salinity stress had an adverse effect on their essential oil content [22].

Nowadays, with development of nanotechnology, the use of nano-fertilizers instead of chemical fertilizers is becoming more popular [23]. Application of micronutrients in nanoscale improves plants growth factors and increases their resistance to salinity stress [24,25]. Various methods such as seed covering, fertilizer supplementation and foliar spraying have been reported for selenium (Se) applications, among which foliar spraying is the most suitable because it can greatly prevent the toxic symptoms of Se accumulation [26,27]. It has been reported that foliar application of Se increases the plant growth and improve the plants tolerance to abiotic stresses [28]. On the other hands, Se improves the plants salinity tolerance by increasing the antioxidant activity of enzymes and strengthening antioxidants and secondary metabolite metabolism [28,29]. In addition, Se application increased the nutritional value and growth parameters of some plants under controlled conditions, indicating that Se is an important micronutrient for plants [29,30,31]. Interestingly, previous research’s reported that selenium nanoparticles (Se-NPs), due to greater surface-to-volume ratio, have better mobility, solubility and bioactivity compared with Se [32,33].

Salinity is one of the main limiting factors in agricultural ecosystems. Due to the high medicinal important and economic value of pineapple mint the current study was conducted to explore the beneficial effects of foliar application of Se-NPs on some morphological and phytochemical traits of the plant under salinity stress. The study was aimed to obtain the best level of this micronutrient concentration for improving pineapple mint quality and quality under salinity stress. 

## 2. Results

### 2.1. Growth Characteristics 

Salinity stress significantly reduced the fresh weight (FW) and plant high (PH) of pineapple mint plants compared with control conditions. Among different salinity levels, 90 mM NaCl exhibited a relatively high reduction of 16.40 and 19.10% in the FW and PH, respectively, compared with non-saline control. However, no significant difference was observed in dry weight (DW) under salinity stress (Table 1). The foliar application of 10 mg L^−^^1^ Se-NPs- on pineapple mint plants displayed a mild increase in FW and DW. Specifically, the foliar spray of 10 mg L^−^^1^ Se-NPs increased FW (by 1.01%), and DW (by 3.80%) over that of Se-NPs-untreated plants under non-saline conditions. However, foliar application of Se-NPs (10 mg L^−^^1^) had no significant effect on PH compared with the normal growth conditions (Table 1). In severe salinity Se-NPs (10 mg L^−^^1^) sprays led to slight increases in DW and PH of 1.36, and 12.16%, respectively (Table 1). Pearson correlation analysis showed a significant positive correlation between dry weight and fresh weight (r = 0.64, *p* value < 0.01) (Table 2).

### 2.2. Relative Water Content and Chlorophyll Index 

The adverse effect of salinity stress on RWC and Chl index contributed to the significant reduction in the RWC and Chl index. Severe salinity (90 mM NaCl) showed a decrease of 18.05 and 3.50% in RWC and Chl index, respectively, in comparison with normal growth conditions (Table 1). This positive effect of 10 mg L^−^^1^ Se-NPs on pineapple mint increased the RWC (by 4.08%), and Chl index (by 1.25%), as compared with non-treated plants grown in normal conditions (Table 1). In severe salinity, Se-NPs (10 mg L^−^^1^) sprays led to slight increases in RWC and Chl index of 9.59, and 12.37%, respectively (Table 1). No significant correlation was observed between RWC and Chl index (Table 2).

### 2.3. Essential Oil Content and Compositions

Essential oil content was ranged among 0.25–0.37 % depending on the Se-NPs treatments and salinity level (Table 1). Application of 10 mg L^−^^1^ Se-NPs levels led to slight increases in essential oil content even under salinity stress. Se-NPs (10 mg L^−^^1^) sprays increased the essential oil content by 25.9% and 17.2% under mild (30 mM) and moderate (90 mM) salinity stress (90 mM NaCl) compared with non-sprayed plants, respectively. 

Based on GC-FID and GC-MS analysis [34], 19 compounds were identified in *M. suaveolens* essential oil (Table 3). In the present study, piperitenone oxide, germacrene D, limonene, octen-3-yl acetate and viridiflorol were identified as the predominant compounds of the species and piperitenone oxide was the most abundant compound (67.7–79.4%) of the essential oils. Foliar application of Se-NPs and different salinity levels significantly affected piperitenone oxide, limonene, jasmone, β-myrcene and viridiflorol contents. The highest (79.4%) and lowest (67.7%) percentages of piperitenone oxide were recorded in the control (no salinity) treatment with 10 mgL^−^^1^ Se-NPs and the mild salinity treatment (10 mM) without Se-NPs application, respectively. The amount of this compound decreased in moderate and mild salinity stresses without Se-NPs treatment compared with the controlled conditions. Mild salinity stress (30 mM NaCl) without Se-NPs- treatment increased the germacrene D content by 78.18% compared with the control condition, but with increasing salinity levels to 60 and 90 mM without Se-NPs treatment, the content of this compound decreased compared with the control condition. The highest and lowest content of limonene were observed at medium salinity (60 mM NaCl) without foliar application of Se-NPs- and severe salinity (90 mM NaCl) without foliar Se-NPs, respectively (Table 4). In fact, increasing salinity stress to a moderate level increased the content of limonene compared with the control condition, however, with increasing the salinity stress to 90 mM, the amount of this compound slightly decreased. With foliar application of 20 mg L^−^^1^ Se-NPs- without salinity stress, the limonene content was increased by 11% compared with control condition. Mild salinity (30 mM) without Se-NPs- treatment increased the amount of viridiflorol, α-Pinene and jasmone compared with the control. The highest and lowest content of viridiflorol were recorded at severe salinity with 20 mg L^−^^1^ Se-NPs- and control treatment (0 mM) with 10 mg L^−^^1^ Se-NPs, respectively. Jasmone had the highest amount in non-saline condition with 20 mg L^−^^1^ Se-NPs- and in severe salinity with 10 mg L^−^^1^ Se-NPs- the lowest amount of jasmone was recorded. Foliar application of 20 mg L^−^^1^ Se-NPs- without salinity stress increased Jasmone content by 48.9 % compared with the control (0 mg L^−^^1^). On the other hand, foliar application of Se-NPs- without salinity stress had a positive effect on the amount of β-myrcene, so that by increasing the concentration of nanoparticles to 20 mg L^−^^1^, the amount of this compound was increased. Salinity at 30 and 60 mM NaCl without foliar application of Se-NPs- increased the amount of this compound compared with the control, but severe stress decreased the amount of β-myrcene. Pearson correlation analysis showed a significant positive correlation between limonene and β-myrcene (r = 0.92, *p* value < 0.01). However, correlation analysis of essential oil components, displayed significant negative correlations between piperitenone oxide and other main components including Limonene, (Z)-Jasmone, (E)-β-Farnesene and β-Myrcene (Table 2). 

## 3. Discussion

In the present study, FW, PH and RWC of pineapple mint decreased with increasing salinity stress. Inhibition of growth and subsequent reduction of biomass are the most common adverse effects of salinity stress, which ultimately leads to a significant reduction in plants yield [25]. The use of selenium can mitigate the effect of salinity stress [35]. The decrease in vegetative growth of pineapple mint under salinity stress may be due to the plant’s inability to absorb water and nutrients due to drought stress caused by salt [36,37,38]. According to previous studies, growth reduction in plants is an adaptive mechanism to cope with stress because it allows the plant to use less cellular metabolic energy for growth and use it more to deal with stress [39]. A study on pineapple mint, showed that the fresh weight of the plant significantly decreased under salinity stress [7]. Furthermore, Kasrati et al. [15] reported that, the fresh weight of pineapple mint significantly decreased as salinity levels were intensified. In salinity treated plants, lack of proper cell turgor, allocation of more synthesized materials to cope with stress, as well as stress escape mechanisms, can prevent the normal development of cells, reducing the plant height. In an experiment performed on three species of mint, including pineapple mint, a decrease in plant height was reported at all salinity levels [7]. Various studies have reported that selenium, especially at low concentrations, reduces the adverse effects of salinity stress by stimulating plant growth [25]. Stimulation of Se-induced growth under various conditions of environmental stress has been linked to increased photosynthesis and activity of antioxidant enzymes.

The relative water content (RWC) of pineapple mint in this study decreased with increasing salinity levels. Under salinity stress, the plants accumulate salt or synthesizes organic compounds such as sugars and amino acids to maintain its osmotic capacity to absorb water and therefore, in such conditions, the RWC is greatly reduced by increasing the salt concentration [40]. Decreased RWC can be affected by reduced root growth and plant weakness in response to salinity [41,42,43,44]. The RWC in soybean *(Glycine max*) decreased under salinity stress [42]. Also, during salinity stress in *Lygeum spartum*, the RWC decreases, which is probably one of the strategies of plant resistance to salinity [44]. Application of selenium increases the amount of antioxidants and reduces the activity of ROS in plants, which in turn increases the stability of cell membranes and improves water potential under stress [44]. 

Chlorophyll index in pineapple mint was significantly different due to foliar application of Se-NPs- under salinity stress. The highest SPAD was recorded in the non-salinity treatments with 20 mg L^−1^ Se-NPs- and the lowest SPAD was recorded in the sever (90 mM NaCl) treatments without foliar application. Decreased SPAD under salinity stress has been attributed to rapid leaf growth [45]. Also, when plants are affected by salinity, thylakoids swelling occurs in the early stages of injury and leads to a decrease in chlorophyll synthesis [46]. In coriander (*Coriandrum sativum* L.) plants selenium significantly increased chlorophyll index under salinity stress. In this study, the highest SPAD was recorded with foliar application of 25 ppm selenium [44,47]. One of the strategies of plants in the face of stress is to increase the production of defense compounds such as secondary metabolites [13,48]. Medicinal plants need adequate levels of micronutrients to grow and produce secondary metabolites. Through foliar application of micronutrients, plant growth can be improved under stress. For instance, it has been reported that the use of micronutrients increased the number of essential oil-secreting glands in mint and thus improved the amount of essential oil [49]. Micronutrient spraying increased the dry matter and components of peppermint (*Mentha piperita* L.) essential oil [50]. Changes in the amount of secondary metabolites in some plants have been reported due to the use of selenium. In salinity stressed coriander (*Coriandrum sativum* L.) plants, selenium foliar application increased the content of γ-terpinene and geranyl acetate compared with the control. Also, high levels of camphene, γ-terpinene and α-terpinolene, D-limonene, β-citronellol, borneol and geraniol were recorded at 50 ppm selenium application [47]. In a study conducted by Tavakoli et al. [51] on *Melissa officinalis*, it was found that the use of 5 μM selenium increased the main constituents of the essential oil, however, the plant is stressed when exposed to high concentrations of selenium. The use of micronutrients such as selenium with increasing the biologically active compounds, improves the medicinal potential of the plants [35]. So far, many and varied combinations of essential oil components of pineapple mint have been reported that this diversity can be due to the diversity of climatic conditions and the region of origin of this species [8,52]. Pineapple mint essential oil contains a high percentage of oxygenated monoterpenes such as piperitone oxide and piperitenone oxide [8,53]. Monoterpenes were the main constituents in pineapple mint essential oil, which was consistent with the results of a study by Aziz et al. [7] on three species of mint, including pennyroyal, peppermint, and pineapple mint. Salinity stress changes the level of essential oil ingredients (increases, decreases and does not change the percentage of different compounds). Changes in the percentage of essential oil constituents depend on the salt concentration used [51]. In a study conducted by Aziz et al. [7] on three species of mint, they found that in peppermint, the percentage of menthol increases with increasing salinity stress, but the percentage of other compounds decreases at the highest salt concentration. Salinity increases the amount of limonene and decreases the amount of carvone in spearmint (*Mentha spicata*) [54]. Piperitenone oxide is one of the main constituents of pineapple mint essential oil (sometimes more than 90%). This oxygenated monoterpene has insecticidal, antiseptic and antioxidant properties. Some data suggest antiviral effects of piperitenone oxide. In addition, it is an analgesic, anti-inflammatory and anti-hypertensive compound [2,55].

## 4. Materials and Methods

### 4.1. Chemicals and Reagents

Chemicals and reagents such as GC grade *n*-hexane, *n*-alkanes (C8-C40 alkanes calibration std.) and authentic standards (for essential oil components) were purchased from Sigma-Aldrich Co. (Saint Louis, MO, USA. The Se-NPs were purchased from Nanosany Corporation Co., Mashhad, Iran.

### 4.2. Experimental Design and Plants Material

A factorial experiment based on a completely randomized design (CRD) with five replications was performed in the greenhouse condition at the University of Maragheh, Maragheh, East Azarbaijan province, Iran (37°30′ N, 46°12′ E, altitude 1477.7 m). Pineapple mint rhizomes were clonally propagated in seed germination tray on 1 November 2019. Then seedlings in four pair-leaves stage were transferred into 5 L pots. The soil mixture used for planting pots was composed of field soil, silt and rotted manure (2:1:1, *v*:*v*). Four levels of water salinity stress including 0 (control), 30, 60 and 90 mM NaCl, and three levels (0, 10 and 20 mg L^−^^1^) of Se-NPs were exerted on *M. suaveolens* in early December. To prevent sudden shock to the plants, salinity stress was applied gradually. Salinity stress continued for about 75 days. Foliar application of Se-NPs- one month after the onset of salinity stress was performed in three stages with an interval of three days [24]. The Se-NPs (Nanosany Corporation, Mashhad, Iran) average particle size ranged from 10 to 45 nm. The specific surface area was 30 to 50 m^2^/g (Figure 1). The plants were harvested at flowering stage (in mid-February, 2020) and some morphological, physiological and phytochemical traits were evaluated.

### 4.3. Growth Parameters 

To measure the FW and DW, the plants in each pot were harvested from five cm above the soil and weight using a sensitive scale. The harvested aerial flowering parts of plants were shade dried at room temperature for two weeks. A Vernier caliper was used to measure the plants height.

### 4.4. Relative Water Content (RWC)

Turner method (Turner, 1986) was used to measure the relative water content (RWC). For this purpose, 10 leaves of each plant were isolated and their FW was measured using a sensitive scale. The leaves were placed in 40 mL of distilled water at 4 °C for 24 h. Then the leaves Turgor weight (TW) was recorded. In the next step, the samples were placed in an oven at 70 °C for 24 h and their DW was recorded.
RWC (%) = [(FW − DW)/(TW − DW)] × 100

### 4.5. Chlorophyll Index

Chlorophyll index was measured using a SPAD-meter (502 Plus Chlorophyll Meter, Konica Minolta, Tokyo, Japan). To determine SPAD value, five randomly leaves from each mint plant were selected. 

### 4.6. Essential Oil Extraction

Essential oil extraction was performed using hydrodistillation method. To extract the pineapple mint essential oil 30 g of dried flowering aerial parts of each sample along with 500 cc of distilled water was placed in a Clevenger apparatus (British pharmacopeia model) and their essential oils were extracted within 3 h (n = 3). The essential oils were dehydrated using anhydrous Na_2_SO_4_ and stored in a dark place at 4 °C until analysis.

### 4.7. Essential Oil Analysis

An Agilent 7990B GC coupled to a 5977A MSD, with HP-5MS capillary column was used for identification of essential oil components. The injector was in split mode (1:35) and its temperature was set at 230 °C. For separation of the essential oil components oven temperature was set at 60 °C for 7 min, then the temperature increased to 240 °C at a rate of 3 °C per min. Carrier gas was Helium. The components were detected in the mass range of 40 to 400 *m*/*z* and electron impact ionization of 70 eV. C8–C40 alkanes calibration std (Supelco, Bellefonte, PA, USA) was used to calculate the retention indices (RI). Identification of components was performed by comparing the calculated RI with those in Adams [34] book, and using the available commercial libraries (WILEY 275 and NIST 17). GC-FID analysis (VF 5MS column; 30 mL, 0.25 mm i.d., 0.50 μm f.t) was performed to calculate the components percentage. The detector temperature was 240 °C. Oven temperature was the same as GC-MS. For components quantification peak area normalization was used [35].

### 4.8. Statistical Analysis

A one-way ANOVA was performed with 95% confidence interval (*p* value ≤ 0.05) using SAS software version 9.1. The means values were compared using least significant difference (LSD) test. Pearson correlation analysis between experimental traits was conducted through SPSS 19 software.

## 5. Conclusions

The current study demonstrated that foliar application of Se-NPs is a useful strategy for improving the plants tolerance to salinity stress. As the salinity increased, FW, PH and RWC decreased. Salinity stress and foliar application of Se-NPs did not have a significant effect on the DW of pineapple mint. Therefore, it can be concluded that this species is somewhat resistant to salinity and can be grown in saline, arid and semiarid regions. Applying the right concentration of Se-NPs increased some of the components of pineapple mint essential oil. Considering that some of the main constituents of essential oil in pineapple mint increased under salinity stress and foliar application of Se-NPs, it can be concluded that stresses, including salinity stress with appropriate concentration of Se-NPs- can affect the production of secondary metabolites in medicinal plants and this feature can be used to engineer the production of secondary compounds and increase the quality of essential oils in this plant and other medicinal plants.

## Figures and Tables

**Figure 1 plants-11-01384-f001:**
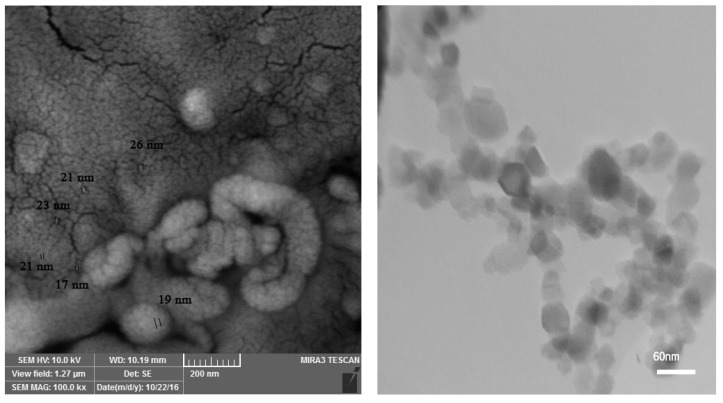
Transmission electron microscopy (TEM; **Left**) and scanning electron microscope (SEM; **Right**) images of Se-NPs.

**Table 1 plants-11-01384-t001:** Effects of foliar application of Se-NPs on the growth parameters and essential oil content of pineapple mint plants under different salinity stress.

	FW	DW	PH	RWC	Chl Index	Essential Oil
	g	g	cm	%		%
Salinity (S)	**	Ns	**	**	ns	Ns
0	86.6 ^a^	18.86	31.66 ^a^	71.02 ^a^	37.75	0.36
30	74.92 ^b^	18.77	28.44 ^b^	65.98 ^ab^	36.74	0.32
60	73.1 ^b^	19.54	28 ^b^	64.3 ^bc^	37.78	0.31
90	72.36 ^b^	19.01	25.61 ^b^	58.2 ^c^	36.42	0.29
Se (NPs)	ns	Ns	ns	ns	ns	Ns
0	77.43	18.64	28.58	63.18	36.88	0.30
10	78.26	19.35	28.25	65.76	37.34	0.34
20	74.54	19.15	28.45	65.05	37.29	0.32
S × NPs	ns	Ns	ns	ns	*	Ns
0 × 0	83.33	17.43	32.66	67.22	36.49 ^bac^	0.35
0 × 10	95.46	20.76	31.83	67.20	36.89 ^bac^	0.36
0 × 20	81.10	18.40	30.50	78.64	39.87 ^a^	0.37
30 × 0	74.50	19.03	28.50	67.18	36.67 ^bac^	0.27
30 × 10	73.86	18.76	28.33	61.63	37.55 ^bac^	0.34
30 × 20	68.73	18.53	28.50	64.09	36 ^bc^	0.35
60 × 0	72.36	19.00	28.50	55.30	39.94 ^a^	0.29
60 × 10	68.23	18.50	25.16	62.34	36.25 ^bc^	0.34
60 × 20	78.70	21.13	30.33	56.96	37.16 ^bac^	0.32
90 × 0	79.63	19.10	24.66	65.57	34.42 ^c^	0.32
90 × 10	75.50	19.36	27.66	71.86	38.68 ^ba^	0.31
90 × 20	69.63	18.56	24.50	60.53	36.15 ^bc^	0.25

Data are means of three independent replications (n = 3). Values in columns with different letters are significantly different at *p* value ≤ 0.05 (least significant difference (LSD) test). * and **, significant difference at 5 and 1%, respectively.

**Table 2 plants-11-01384-t002:** Pearson’s correlation coefficients among grows parameters, essential oil content and compositions of pineapple mint under different salinity stress and selenium nanoparticle applications.

	Fresh Weight	Dry Weight	Plant Height	RWC	Chl Index	Essential Oil Content	Piperitenone Oxide	β-Myrcene	Limonene	(Z)-Jasmone	(E)-β-Farnesene	Germacrene D	Viridiflorol
Fresh weight	1.00												
Dry weight	0.64 **	1.00											
Plant height	0.31	0.07	1.00										
RWC	0.31	0.02	0.14	1.00									
Chl index	0.12	0.18	0.27	0.07	1.00								
Essential oil content	0.21	0.01	0.20	0.29	0.02	1.00							
Piperitenone oxide	0.29	0.03	−0.05	0.06	−0.23	0.19	1.00						
β-Myrcene	−0.36 *	0.09	−0.14	−0.39 *	0.17	−0.42 *	−0.78 **	1.00					
Limonene	−0.23	0.01	0.04	−0.21	0.29	−0.366 *	−0.86 **	0.92 **	1.00				
(Z)-Jasmone	−0.17	−0.07	0.07	−0.07	0.15	−0.11	−0.68 **	0.73 **	0.78 **	1.00			
(E)-β-Farnesene	−0.22	0.05	0.06	−0.05	0.13	−0.16	−0.90 **	0.75 **	0.77 **	0.70 **	1.00		
Germacrene D	−0.06	−0.05	0.27	0.05	0.17	0.01	−0.83 **	0.62 **	0.75 **	0.79 **	0.89 **	1.00	
Viridiflorol	−0.31	−0.09	−0.04	0.02	0.13	−0.13	−0.72 **	0.30	0.365 *	0.13	0.55 **	0.38 *	1.00

* and **, significant difference at 5 and 1%, respectively.

**Table 3 plants-11-01384-t003:** Amount of pineapple mint essential oil compounds under different levels of salinity and foliar application of Se-NPs.

Constituents	RI *	Treatments											
		0 (mM)			30 (mM)			60 (mM)			90 (mM)		
		0	10	20	0	10	20	0	10	20	0	10	20
α-Pinene	930	0.53	0.53	0.56	0.73	0.68	0.75	0.73	0.62	0.68	0.56	0.64	0.62
Sabinene	969	0.29	0.29	0.32	0.44	0.39	0.42	0.44	0.36	0.39	0.36	0.39	0.4
*β*-Pinene	972	0.67	0.69	0.75	0.97	0.86	0.93	0.97	0.82	0.89	0.80	0.89	0.89
*β*-Myrcene	989	0.51	0.52	0.59	0.82	0.72	0.74	0.84	0.62	0.65	0.55	0.52	0.69
n-Decane	998	0.07	0.03	0.03	0.03	0.06	0	0	0	0	0	0	0
Limonene	1025	2.75	2.45	3.09	4.01	3.44	3.32	4.09	2.6	2.61	2.44	2.54	2.98
(Z)-β-Ocimene	1035	0.17	0.16	0.21	0.25	0.22	0.22	0.23	0.09	0.13	0.09	0.12	0.13
*n*-Undecane	1098	0.08	0.06	0.09	0.14	0.13	0.12	0.13	0.05	0.05	0.09	0.11	0.05
1-Octen-3-yl acetate	1111	1.89	1.77	2.45	3.0	2.75	2.75	3.01	2.59	2.57	2.69	3.25	2.66
Borneol	1160	0.1	0.17	0.37	0.43	0.25	0.32	0.7	0.54	0.56	0.75	0.84	0.65
*n*-Dodecane	1198	0.53	0.74	0.8	1.25	1.24	1.69	0.79	1.11	1.08	1.16	1.19	1.02
Piperitenone oxide	1368	76.69	79.45	72.97	67.74	69.26	67.85	70.47	76.85	77.39	78.69	75.72	76.35
β-Bourbonene	1380	0.22	0.22	0.27	0.32	0.33	0.32	0.29	0.24	0.34	0.19	0.12	0.39
β-Elemene	1388	0.37	0.33	0.48	0.58	0.83	0.91	0.66	0.65	0.6	0.55	0.68	0.63
(Z)-Jasmone	1394	0.92	0.9	1.37	1.34	1.15	1.26	1.32	0.99	0.95	0.89	0.6	0.01
trans-Caryophyllene	1413	0.52	0.49	0.65	0.72	0.75	0.86	0.62	0.52	0.48	0.43	0.39	0.52
(E)-β-Farnesene	1455	0.99	0.9	1.22	1.37	1.41	1.53	1.17	1.02	1.0	0.91	0.88	1.07
Germacrene D	1476	6.65	5.4	7.48	8.0	8.06	8.65	6.81	4.99	4.2	3.37	3.17	4.49
Viridiflorol	1585	1.69	1.34	1.78	2.11	2.19	2.35	1.78	1.73	1.54	1.67	2.26	1.73

* The calculated retention index.

**Table 4 plants-11-01384-t004:** Effects of foliar application of Se-NPs on the Essential oil components of pineapple mint plants under different salinity stress.

Treatments		Piperitenone Oxide	Limonene	Viridiflorol	Jasmone	*β*-Myrcene
Salinity (mM)	Se-NPs-(mg L^−1^)					
0	0	76.69 ^ba^	2.75 ^dc^	1.69 ^edc^	0.92 ^c^	0.51 ^e^
	10	79.45 ^a^	2.45 ^d^	1.34 ^e^	0.9 ^c^	0.52 ^e^
	20	72.97 ^bc^	3.09 ^dc^	1.78 ^bdc^	1.37 ^a^	0.59 ^ecd^
30	0	67.74 ^d^	4.01 ^ba^	2.11 ^bac^	1.34 ^a^	0.82 ^a^
	10	69.26 ^dc^	3.44 ^bac^	2.19 ^ba^	1.15 ^ba^	0.72 ^ba^
	20	67.85 ^d^	3.32 ^bc^	2.35 ^a^	1.26 ^ba^	0.74 ^ba^
60	0	70.47 ^dc^	4.09 ^a^	1.78 ^bdc^	1.32 ^a^	0.84 ^a^
	10	76.85 ^ba^	2.6 ^d^	1.73 ^edc^	0.99 ^c^	0.62 ^becd^
	20	77.39 ^a^	2.61 ^d^	1.54 ^ed^	0.95 ^c^	0.65 ^bcd^
90	0	78.69 ^a^	2.44 ^d^	1.67 ^ed^	0.89 ^c^	0.55 ^ed^
	10	75.72 ^ba^	2.54 ^d^	2.26 ^a^	0.6 ^d^	0.52 ^e^
	20	76.35 ^ba^	2.98 ^dc^	1.73 ^edc^	1 ^bc^	0.69 ^bc^

Data are means of three independent replications (n = 3). Values in columns with different letters are significantly different at *p* value ≤ 0.05 (least significant difference (LSD) test).

## Data Availability

Not applicable.

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
