# Peer review of "Selenium Nanoparticles (Se-NPs) Alleviates Salinity Damages and Improves Phytochemical Characteristics of Pineapple Mint (*Mentha suaveolens* Ehrh.)"

_plants, 2022, doi:10.3390/plants11101384_

Round 1
Reviewer 1 Report
- The abstract has many results; no data on working methods. What secondary metabolites were analyzed?
- Please complete with some morphological characters specific to mint species.
- Line 279 - begins with a capital letter.
- When were they harvested and what was harvested from Mentha suaveolens? At what stage was M. suaveolens growing?
- By what process was the plant material dried?
- Please specify the type of essential oil (by what method?), As well as the vegetable materials used to extract the oil.
- A section with the chemicals used was not mentioned. Have NPs been purchased? Were the nanoparticles synthesized with M. suaveolens extract?
- How was Figure 1 obtained? The techniques mentioned in the legend Fig. 1. (Transmission electron microscopy (TEM; Left) and scanning electron micro- 264 scope (SEM; Right) images of Se-NPs).
Author Response
Reviewer #1
The abstract has many results; no data on working methods. What secondary metabolites were analyzed?
Our reply# This section was revised accordingly and we mentioned that “Based on GC-FID and GC-MS analysis, 19 compounds were identified in pineapple mint essential oil.” We mentioned to the essential oil components as the secondary metabolites as well.
Please complete with some morphological characters specific to mint species.
Our reply# Done. As a result of interspecific hybridization, the genus Mentha shows a vast range of morphological and phytochemical variability. Also, this sentence was added to the introduction “Pineapple mint is one of the variegated cultivars (cv. variegata) of M. suaveolens possessing bumpy and hairy leaves usually surrounded with white margins.”
Line 279 - begins with a capital letter.
Our reply# Done.
When were they harvested and what was harvested from Mentha suaveolens? At what stage was M. suaveolens growing?
Our reply# The plants were harvested at flowering stage (in mid-February, 2020) and some morpho-logical, physiological and phytochemical traits were evaluated.
By what process was the plant material dried?
Our reply# We mentioned that “The harvested aerial flowering parts of plants were shade dried at room temperature for two weeks”
Please specify the type of essential oil (by what method?), As well as the vegetable materials used to extract the oil.
Our reply# Done. Essential oil extraction was performed using hydrodistillation method. To extract the pineapple mint essential oil 30 g of dried flowering aerial parts of each sample along with 500 cc of distilled water was placed in a Clevenger apparatus.
A section with the chemicals used was not mentioned. Have NPs been purchased? Were the nanoparticles synthesized with M. suaveolens extract?
Our reply# Section was added. The Se-NPs were purchased from Nanosany Corporation Co., Mashhad, Iran.
How was Figure 1 obtained? The techniques mentioned in the legend Fig. 1. (Transmission electron microscopy (TEM; Left) and scanning electron micro- 264 scope (SEM; Right) images of Se-NPs).
Our reply# the company provided TEM and SEM

Author Response
Reviewer 2.
The paper titled “Selenium nanoparticles (Se-NPs) alleviates salinity damages and improves phytochemical characteristics of pineapple mint (Mentha suaveolens Ehrh.)”, represents an interesting work and is well written. In my opinion, after a few minor changes can be considered for publication in this Journal.
Our reply# Thanks for your positive comments
I recommend to the authors to read all the text carefully; some mistakes are present. For example:
- Page 1, line 34: undber 60 Mm NaCl increased. Correct the error
Our reply# done
- Modified the formule, Na2SO4 (pag. 9, line 286)
Our reply# done
3) Pag. 9, 4.6. Essential oil analysis. There are repletion in this section, as the temperature 240°C.
Our reply# As we have used the different instruments (GC and GC-MS) for essential oil analysis we have used this temperature for detector and transfer line and injector temperature. They are different.
4) Standardize the P value in the text, as you reported in the Statistical analysis section. I also recommend that you check the English form.
Our reply# revised
You could insert bibliographic references more recent (last 5 years). Read and eventually quote this work: De Bruno, Alessandra, Amalia Piscopo, Francesco Cordopatri, Marco Poiana, and Rocco Mafrica. 2020. "Effect of Agronomical and Technological Treatments to Obtain Selenium-Fortified Table Olives" Agriculture 10, no. 7: 284. https://doi.org/10.3390/agriculture10070284
Our reply# We revised the references list and added some new references.

Round 2
Reviewer 1 Report
The authors completed and resolved the requirements. Thank you very much!